# Relationship of Carbohydrate Intake during a Single-Stage One-Day Ultra-Trail Race with Fatigue Outcomes and Gastrointestinal Problems: A Systematic Review

**DOI:** 10.3390/ijerph18115737

**Published:** 2021-05-27

**Authors:** Soledad Arribalzaga, Aitor Viribay, Julio Calleja-González, Diego Fernández-Lázaro, Arkaitz Castañeda-Babarro, Juan Mielgo-Ayuso

**Affiliations:** 1Institute of Biomedicine (IBIOMED), Physiotherapy Department, University of Leon, 24071 Leon, Spain; marisolarribal@gmail.com; 2Glut4Science, Physiology, Nutrition and Sport, 01004 Vitoria-Gasteiz, Spain; aitor@glut4science.com; 3Department of Physical Education and Sport, Faculty of Education and Sport, University of the Basque Country (UPV/EHU), 01007 Vitoria, Spain; julio.calleja.gonzalez@gmail.com; 4Department of Cellular Biology, Histology and Pharmacology, Faculty of Health Sciences, University of Valladolid, 42003 Soria, Spain; diego.fernandez.lazaro@uva.es; 5Neurobiology Research Group, Faculty of Medicine, University of Valladolid, 47005 Valladolid, Spain; 6Health, Physical Activity and Sports Science Laboratory, Department of Physical Activity and Sports, Faculty of Psychology and Education, University of Deusto, 48007 Bizkaia, Spain; arkaitz.castaneda@deusto.es; 7Department of Health Sciences, Faculty of Health Sciences, University of Burgos, 09001 Burgos, Spain

**Keywords:** marathon trail, recovery, food intake, ultramarathon, carbohydrates, GI symptoms

## Abstract

Due to the high metabolic and physical demands in single-stage one-day ultra-trail (SOUT) races, athletes should be properly prepared in both physical and nutritional aspects in order to delay fatigue and avoid associated difficulties. However, high carbohydrate (CHO) intake would seem to increase gastrointestinal (GI) problems. The main purpose of this systematic review was to evaluate CHO intake during SOUT events as well as its relationship with fatigue (in terms of internal exercise load, exercise-induced muscle damage (EIMD) and post-exercise recovery) and GI problems. A structured search was carried out in accordance with PRISMA guidelines in the following: Web of Science, Cochrane Library and Scopus databases up to 16 March 2021. After conducting the search and applying the inclusion/exclusion criteria, eight articles in total were included in this systematic review, in all of which CHO intake involved gels, energy bars and sports drinks. Two studies associated higher CHO consumption (120 g/h) with an improvement in internal exercise load. Likewise, these studies observed that SOUT runners whose intake was 120 g/h could benefit by limiting the EIMD observed by CK (creatine kinase), LDH (lactate dehydrogenase) and GOT (aspartate aminotransferase), and also improve recovery of high intensity running capacity 24 h after a trail marathon. In six studies, athletes had GI symptoms between 65–82%. In summary, most of the runners did not meet CHO intake standard recommendations for SOUT events (90 g/h), while athletes who consumed more CHO experienced a reduction in internal exercise load, limited EIMD and improvement in post-exercise recovery. Conversely, the GI symptoms were recurrent in SOUT athletes depending on altitude, environmental conditions and running speed. Therefore, a high CHO intake during SOUT events is important to delay fatigue and avoid GI complications, and to ensure high intake, it is necessary to implement intestinal training protocols.

## 1. Introduction

Participation in single-stage one-day ultra-trail (SOUT) races has increased in recent years [1]. These types of events (ultra-endurance races) are defined as competitions of more than 4 h duration but less than 24 h, where each participant must run as many kilometers as possible over 24 h [2,3,4,5,6,7,8,9,10]. Due to the high metabolic and physical demands in these types of events, athletes should be properly prepared in both physical and nutritional aspects, in order to delay fatigue and avoid associated difficulties [6,10,11,12,13]. Fatigue is classified as proximal (central fatigue) or distal (peripheral fatigue) to the neuromuscular function [14]. On the one hand, the first implies a reduction in activation and reflects the inability to recruit all the motor units needed to maximize strength. On the other hand, peripheral fatigue could affect three main components [15]:(1)action potential transmission along the sarcolemma;(2)excitation–contraction coupling;(3)actin-myosin interaction.

The fatigue generated in these types of events and its impact on internal load during the race is associated with a reduction in glycogen stores and exercise-induced muscle damage (EIMD). EIMD could affect performance by a temporary reduction in muscle strength, increasing muscle pain and swelling, and even lead to failure to complete the event [8,16]. In this context, medical complications related to dietary-nutritional intake such as gastrointestinal (GI) problems, hyponatremia, dehydration and appetite suppression, as well as difficulty in carrying food and/or supplements, are evident. These are the main reasons why suitable nutrition [17] is a key factor for optimal performance [11].

In prolonged strenuous exercise, such as in SOUT events, nutritional planning and suitable CHO intake during the event reduce fatigue time [18,19], and consequently, high CHO intake rates were significantly correlated with faster finishing times [20]. Moreover, although it seems there is no linear dose–response to CHO ingestion [21], an optimal CHO intake could maintain plasma glucose and CHO oxidation rates [22] and augment exercise performance via multiple mechanisms, consisting of muscle glycogen sparing [23], and liver glycogen sparing [24] deemed necessary for optimal sports performance [11,13,25,26,27]. As such, different international nutrition societies recommend a 90 g/h intake of CHO in exercises of more than 3 h duration (with a combination of CHO that use different absorption transporters such as glucose or fructose) to improve athletic performance by gastric emptying, maximize their oxidation and reduce possible GI discomfort via suitable gut training [13,26,27,28]. Despite these recommendations, new protocols involving 120 g/h CHO intake during a mountain marathon might reduce the impact of exercise internal load [4], limit EIMD and neuromuscular fatigue, and improve recovery of a high intensity run capacity 24 h afterward [2].

Despite CHO intake recommendations, the vast majority of ultra-endurance athletes consume smaller amounts of CHO compared to macronutrient recommendations during SOUT competitions [5,6,7,8,9,20,29,30,31,32,33]. The following factors may encourage athletes not to meet CHO needs during SOUT races: unfavorable geographical and environmental conditions; race experience, [33]; running speed or intensity [5]; limited food options at checkpoints (i.e., frequency or type of food) [9]. Moreover, GI problems have become one of the main problems faced by ultra-endurance runners [6], and such complaints may be of varying severity, although symptoms may include nausea, vomiting, abdominal angina, and bloody diarrhea [34]. Between 30% and 90% of these athletes have experienced some types of GI problem while competing [6,8,20,32,35,36]. While the causes of GI symptoms during prolonged exercise are multifactorial [32,37,38], CHO contained in foods and beverages could intensify these situations [39], and so GI problems could be another reason why SOUT athletes do not adhere to nutritional recommendations [5,6,8,32,35,38,40,41]. However, GI problems caused by CHO intake could be reduced by intestinal training and suitable hydration and nutrition strategy [8,42]. In many cases, these problems not only have negative effects on performance but also an impact on subsequent recovery [34,43,44]. 

Although some extensive reviews provide general information about CHO intake and its relationship with GI symptoms [37,45], to the best of the author’s knowledge, none of them have explored the relationship between CHO intake with fatigue parameters and GI problems in detail. Therefore, the main aim of this systematic review was to collect the information related to CHO intake by SOUT athletes, as well as its relationship with fatigue (in terms of internal exercise load, EIMD, post-exercise recovery) and GI problems. 

## 2. Methods

### 2.1. Literature Search Strategies

This systematic review was carried out in accordance with the guidelines of the Preferred Reporting Items for Systematic Review and Meta-Analyses (PRISMA^®^) [46] and the PICOS model [47] for definition of the following inclusion criteria: P (Population): “endurance and ultra-endurance runners”, I (Intervention): “CHO intake during SOUT”, C (Comparators): “international recommendations for CHO intake at events lasting more than 3 h”, O (Outcomes): “CHO intake, post-exercise recovery, internal load and GI problems“, S (Study design): “any type of design”. A systematic search of current scientific literature was undertaken to find up-to-date information about CHO intake consumed by ultra-trail athletes who compete in events of more than 4 h duration during SOUT races, as well as its relationship with exercise load, EIMD and GI problems.

A structured search was conducted of the Medline (PubMed), Scopus and Web of Science (WOS) database that includes other databases such as BCI, BIOSIS, CCC, DIIDW, INSPEC, KJD, RSCI and SCIELO, all of which are of high quality and guarantee suitable bibliographic support for this systematic review completed on 16 March 2021. The keywords provided in the full article were used in accordance with the following Boolean search equation: (“marathon running”[MeSH Terms] OR (“marathon”[All Fields] AND “running”[All Fields]) OR “marathon running”[All Fields] OR “marathon”[All Fields] OR “marathons”[All Fields] OR “marathoner”[All Fields] OR “marathoners”[All Fields] OR (“ultramarathon”[All Fields] OR “ultramarathoners”[All Fields] OR “ultramarathons”[All Fields]) OR “ultra-endurance”[All Fields] OR “ultra-trail”[All Fields]) AND (“food intake”[All Fields] OR “fluid intake”[All Fields] OR (“carbohydrate s”[All Fields] OR “carbohydrated”[All Fields] OR “carbohydrates”[MeSH Terms] OR “carbohydrates”[All Fields] OR “carbohydrate”[All Fields])) AND (“post-exercise recovery”[All Fields] OR (“recoveries”[All Fields] OR “recovery”[All Fields]) OR “gastrointestinal problems”[All Fields] OR “gastrointestinal events”[All Fields] OR “exercise induced muscle damage”[All Fields] OR “muscle damage”[All Fields] OR (“fatiguability”[All Fields] OR “fatiguable”[All Fields] OR “fatigue”[MeSH Terms] OR “fatigue”[All Fields] OR “fatigued”[All Fields] OR “fatigues”[All Fields] OR “fatiguing”[All Fields] OR “fatigueability”[All Fields]).

Relevant articles were also obtained using this equation by applying the snowball strategy, with all titles and abstracts from the search being cross-referenced to identify duplicates and any potentially missing studies. Titles and abstracts were selected for further review of the full text. The search for published studies was carried out independently by two authors (S.A. and J.M.A.), and any disagreements about all outcomes were resolved through discussion (J.C.G.).

### 2.2. Inclusion and Exclusion Criteria

The following inclusion criteria were applied in selecting studies for the articles found in the search: (I) studies that were carried out during a SOUT race; (II) reporting on CHO intake (quantity and/or time of intake); (III) inclusion of post-exercise recovery, internal load information and/or description of GI complications associated with CHO intake during SOUT events; (IV) inclusion of those performed on any number or type of athlete regardless of category, training status or gender; (V) languages restricted to English, German, French, Italian, Spanish and Portuguese.

The following exclusion criteria were applied to the experimental protocols of this systematic review: (I) ultra-trail events that exceed 24 h or ultra-endurance races on the road, (II) those performed by bike or some other specialty that is not agreed; (III) absence of information about CHO intake; (IV) that the study was conducted on participants with a pathological condition (IV) those that exclusively describe other nutrients during a marathon, ultra-trail or ultra-marathon. There were no filters applied to athletes’ level, race, gender, ethnicity, or age in order to increase the analytical power of the analysis itself. 

### 2.3. Study Selection

Once the inclusion/exclusion criteria had been applied to each study, data on study source (including authors and year of publication), study design, supplement administration (dose and timing), sample size, participant characteristics (level, race and gender) and final outcomes of the interventions were mined independently by two authors (S.A. and J.M.A.) using a spreadsheet (Microsoft Inc, Seattle, WA, USA). Subsequently, disagreements were resolved through discussion until a consensus was reached or there was third-party adjudication (J.C.G.).

### 2.4. Outcome Measures

The literature studied information related to the CHO intake by ultra-endurance athletes during SOUT, as well as its effect on post-exercise recovery, exercise load and/or GI problems, using several outcome variables such as CHO intake (amount, type of CHO and nutritional strategies), fatigue (exercise internal load, EIMD, recovery) and upper and lower GI symptoms. The results could have been influenced by the type of event, amount of each supplement and duration of the intervention (races). Participant’s characteristics, such as age, gender, ethnicity, body composition, training level, differences in training, nutrition and health status and ethnicity, may also have influenced the results.

### 2.5. Quality Assessment of the Experiments

Methodological quality and risk of bias were assessed in accordance with qualitative studies via a McMaster Critical Review Form [48] by two authors independently (J.M.A. and S.A.), with disagreements being resolved by third-party evaluation (A.V.). There were 16 evaluated items (purpose, literature review, study design, blinding, sample description, sample size, ethics and consent, validity of outcomes, reliability of outcomes, intervention description, statistical significance, statistical analysis, clinical importance, conclusions, clinical implications, study limitations), which were rated as “1” if the criteria were fully met, “0” if they were not completely met, or “NR” in cases where information was not reported. Study scores were as follows: ≤8 points indicated poor quality; 9–10 points acceptable quality; 11–12 points good quality; 13–14 points very good quality and ≥15 points excellent quality. As such, 1 out of 8 studies included in this systematic review evidenced poor quality, 1 acceptable quality, 5 studies good quality and 1 very good quality (Table 1).

## 3. Results

### 3.1. Main Search

The initial literature search provided a total number of 93 relevant articles related to the descriptors selected, although only 8 articles [2,3,4,5,6,8,29,33] met all the inclusion criteria (Figure 1). The number of articles and reasons for exclusion were as follows: Twenty-five of them were associated with alternative diets (diets-ketosis, high in fat, low in CHO, supplements, and antioxidants), 10 studies with training and performance, 9 referred to pathologies, 5 to reviews, 29 failed to incorporate some of the inclusion criteria (24 referred to other sports and 5 analyzed the nutritional knowledge of athletes). Moreover, 9 studies were excluded because they focused on other outcomes.

### 3.2. Characteristics of SOUT Events

Among the 8 studies selected, 6 were ultra-trail races held in Europe (1 in Spain, 1 in France, 2 in Holland, 1 in Switzerland, 1 in Scotland) [3,5,6,8,29,33] and 3 were in marathon trail races [2,4,29] held in Spain (Table 2).

The distances traveled in SOUT events were between 44 km covered by the “Tour des Dents du Midi” race in Switzerland [33] and 272 km covered by the World Championship held in Albi (France) [3]. However, 3 of the 9 SOUT events involved over 100 km [5,6,7,8,9,10,29], while the Glenmore 24 Trail Race event in Scotland was computed by the maximum distance traveled in 24 h, the range of which was 122–208 km [8]. One of the 6 events involved a trail marathon [29]. The cumulative elevation gain of the ultra-trail tests of more than 100 km ranged between 342 m in the Glenmore 24 Trail Race in Scotland [8] and 4448 m in the Ultra Mallorca Serra de Tramuntana in Spain [29]. Aditionally, that of ultra-trail races of less than 100 km ranged between 4448 m in the Transvulcania, Spain [29] and 2521 m of in the 67 km of the Ultra Mallorca Serra de Tramuntana, Spain [29]. The temperature at which the ultra-trail events were carried out was between 2.2 °C over 120 km in the Sixty of Texel [6] and 35.1 °C in the 272 km of the World Championship held in France [3]. 

As for the environmental conditions of the mountain marathon, this took place within a temperature range of 11.0–21.6 °C in the Ultra Mallorca Serra de Tramuntana, Spain, with an ascent of 1424 m [29]. It should also be taken into consideration that 1 study [29] covered the events in 2 categories: ultra-trail and trail marathon.

### 3.3. Characteristics of SOUT Runners

417 SOUT runners in total were included in this systematic review. Specifically, participants comprised 365 male and 52 female runners (Table 2), while participants were only men in 3 of the studies [2,4,5], and the remaining 5 studies included athletes of both genders [3,6,8,29,33]. Furthermore, the total number of runners included in the different ultra-trail events was 314 (271 men and 40 women) aged between 35.2 ± 8.4 years in the study by Martínez et al. [29] and 47 ± 6 years in that by Wardenaar et al., 2018 (22). The latter was a study carried out in the Ultra Mallorca Serra de Tramuntana (Spain) where maximum participation was recorded with 109 athletes (98 of whom were men) (35.7 ± 7.9 years) [29], while the one with the least number of participants was that described by Waardenaar et al. [5]. The total number of runners included in the marathon trail studies was 143 (91 men and 12 women) with an age of 36.6 ± 8 years [29], while the lowest number of participants was 20 [2,4], aged LOW:37.8 ± 9.4; MED: 37.2 ± 5.4; HIGH: 38.0 ± 6.8.

### 3.4. Carbohydrate Intake

The CHO amount ingested during SOUT events ranged between 22.1 g/h in a 120 km race [5] and a maximum of 126 g/h ingested by a runner in a 160 km race [8] (Table 3). In the study by Wardenaar et al. [5], the highest CHO intake was 90.6 ± 38.2 g/h, corresponding to the 75–90 km section, while in the marathon trail the CHO intake was between 205.2 ± 81.2 g (the equivalent of (33.4 ± 13.5 g/h) [29]) and 120 g/h [2,4]. In terms of the type of CHO consumed, this is represented by solid, semi-solid and liquid foods [5], solid foods being fruit [3,5,9,29], gummies [29], pasta [29] and energy bars [5,29], as shown in Table 3. For its part, the liquid intake comprised chocolate milk [5], cola drinks [5,29] and commercial energy drinks in three studies [5,29,33]. One study described the type of CHO, but did not specify the food consumed by athletes [8]. Similarly, all studies allowed their athletes to consume the usual intake [5,6,8,29,33]. 

On the other hand, only three studies [2,3,4] provided nutritional instructions to their athletes for either the previous weeks and/or for ingestion during the race (Table 3). Likewise, one of these studies [3] allowed runners to modify the programmed intake during the race. It should be noted that among the studies on nutritional strategy, a recommendation of 90 g/h of CHO was not necessarily suggested, although 30–50 g/h was used as a target [3]. Despite not having provided athletes with a nutritional strategy, the percentage number in the study that followed recommendations was 22% (21.1% men and 12.5% women) [6] and 69% (100% male) in a further two studies [2,4]. Other relevant points noted in studies referred to the “ad libitum intake”, insofar as athletes were unaware of what was recommended for exercises of this duration and/or of the requirement [6], and so underestimated CHO consumption. In two studies [2,4] athletes had to ingest the amount of CHO determined by the group they were in. In addition, in these articles, an inclusion criterion was that athletes should carry out nutritional training assigned by their professional.

On the other hand, the fluid amount ingested during SOUT events ranged between 274 mL/h in a 120 km race [3] and a maximum of 765 mL/h ingested by a runner in a 160 km race [5]. Two of the studies [2,4] that analyzed a marathon reported an “ad libitum intake” by participants and the last an intake of 459 ± 256 mL/h [29]. Four studies [3,5,29,33] reported intake being maintained close to fluid recommendations of 0.4 to 0.8 L/h [28].

### 3.5. Fatigue Outcomes

Two research articles [2,4] from the eight studies analyzed in this systematic review described internal exercise load, EIMD markers and post-exercise recovery outcomes (Table 4). In terms of exercise internal load and the evaluation of EIMD markers [4] and post-exercise recovery [2], the groups (HIGH or EXP), that consumed the largest amount of CHO per hour evidenced a lower exercise internal load level and EIMD markers (CK, LDK and GOT) after the race. Likewise, the HIGH or EXP groups evidenced better post-exercise recovery as observed by jumps and high intensity running capacity.

### 3.6. GI Symptoms

A common problem among endurance athletes is GI complications [26]. GI distress can be divided into upper: reflux, heartburn, belching, bloating, stomach cramps, vomiting, and nausea, and lower GI complaints may be: intestinal cramp, flatulence, urge to defecate, side ache, abdominal pain, loose stool, diarrhea and intestinal bleeding. The means used to measure GI symptoms was through surveys [5,6] with a nine-point scale (“no problem at all” to “the worst it has ever been”) or were consulted for complications at checkpoints throughout the race and at the end of the race [2,3,4,8]. 

Six [2,3,4,5,6,8] of the nine studies analyzed in this systematic review described GI problems (Table 5). Wardenaar et al. [6] and Costa et al. [8] showed an incidence of GI distress of 82% and 65% respectively, with two of them categorizing GI symptoms into upper and lower GI problems [5,6] The most recurrent upper GI symptoms include: nausea, heartburn, reflux, inflammation and stomach pain [6], and flatulence and diarrhea in the case of lower symptoms [5,6]. Another factor in GI symptoms could be high fiber consumption [6], and apart from nutritional factors, others such as warm environmental conditions, duration and/or intensity of exercise are also described [6]. Although some studies confirm the relationship between intake while competing [5] and GI symptoms, another study did not find this link [8]. In this sense, the alternatives proposed that favor CHO intake were via intestinal adaptation training [2,4,8].

## 4. Discussion

The main aim of this systematic review was to synthesize the information about CHO intake by runners during SOUT events (ultra and marathon trails), the influence of internal exercise load, EIMD markers and post-exercise recovery process on fatigue, and/or the appearance of GI problems. The main results showed that ultra-trail athletes did not adhere to updated recommendations of 90 g/h of CHO during SOUT races [8,28,49]. On the other hand, runners who consumed more CHO reported a low internal exercise load level, minimized EIMD and improved post-exercise recovery outcomes. Likewise, 45 to 75% of athletes evidenced GI distress depending on altitude and environmental conditions [5] and running speed [6].

### 4.1. CHO Intake by Athletes during SOUT Events

The relative contribution of each macronutrient during an ultra-endurance race depends on previous energetic status, exercise intensity and duration and the nutrition strategy pursued prior to and during exercise [13], based on individual and personalized needs [17,26]. In particular, CHO is an important source of energy, although stores in the human body are limited, with muscle and hepatic glycogen being the most important storage areas [19]. For this reason, athletes must continuously ensure intake of this nutrient [12,50], whereby an individualized nutritional strategy may be developed that aims to deliver CHO to the working muscle at a rate that is dependent on absolute exercise intensity as well as the duration of the event [13]. Thus, international consensus establishes the recommendation to be 90 g/h when the activity exceeds 3 h, as well as the appropriate ratio of the type of CHO (glucose/fructose in 2:1 ratio) and previous gut training [8,28,49]. Importantly, ingesting different types of CHO means that absorption is carried out by several intestinal apical transporters (SGLT-1 for glucose and GLUT-5 for fructose) and thus, the ability to achieve maximum absorption and oxidation improves [26]. Conversely, it has been observed that the CHO oxidation rate can reach maximum values of 1.6 g/min when 100 g/h of glucose and fructose combination were ingested during cycling, thus reducing endogenous CHO oxidation [51]. Moreover, consuming glucose in combination with fructose resulted in 55% greater exogenous CHO oxidation rates, compared to ingesting glucose alone [52]. As such, high consumption of CHO with an appropriate composition during long-term exercises allows optimal blood glucose levels to be maintained, glycogen stores preserved, glycolysis enhanced, the efficiency of blood lactate production improved, and better sports performance attained [13,26,53,54]. 

On the other hand, the difficulties that the studies included in this work expose is the fact of not corresponding to updated recommendations during SOUT races linked to appetite suppression [8,10,33], lack of food education [8], difficulty in eating while running [33], factors that motivate food selection include taste [55], convenience, nutrition knowledge and beliefs, GI discomfort [5,8], flavors and tastes preferred by the athlete and the logistics associated with the race such as the great distance among the supply stations or the lack of variety of food in them [8]. Some authors, recommend the use of some nutritional strategies supervised by a professional dietitian or nutritionist or bringing their own food in order to increase food intake [37], although the 90 g/h goal is not always achieved. Accordingly, only two studies used a strategy that met the recommendation of CHO intake (90 g/h) and especially when (120 g/h) at least in one of the experimental groups [2,4]. In particular, a single runner who ingested 108 g/h during the 120 km race was able to adhere to recommendations without professional supervision [6]. However, it may be considered that in some cases the intake used for the strategy was lower than the recommendations (30–50 g/h) [3]. Studies that did not pursue strategies sought to ensure that athletes consumed the same as in training [5,6,8,29,33], except for two studies in which athletes had to pursue an intestinal strategy for at least two weeks [2,4]. 

Regarding the type of CHO food source, Jeukendrup [53,54] pointed out the importance of varying textures in order to avoid flavor fatigue. Along these lines, glucose transporter SGLT-1 is upregulated by molecular mechanism (mediated by enteric neurons) in response not only to glucose or galactose, but also to fructose, sucralose and others [42,56]. This regulation depends only on the bioavailability of these substrates, but not on their metabolism [42]. As the presence of some added sugars, sweeteners and/or flavors in foods can also upregulate glucose transport via SGLT-1 upregulation, flavor and food composition also have important implications through physiological mechanisms [42]. Moreover, one study highlighted the preference of athletes for solid foods as a source of CHO [29]. Thus, they gave a greater feeling of satiety, although some athletes preferred a sports drink, arguing that from a practical standpoint they can hydrate and ingest CHO at the same time without their having to load up with snacks or other solid foods [10]. Similarly, in terms of the consistency of food, Pfeiffer et al. [57] concluded that the average and maximum exogenous CHO oxidation rate was similar when athletes consumed a mixture of glucose + fructose in solid form (energy bar) or in a liquid state (sports drink). In other words, the gastric emptying rates were not substantially modified by the consistency of CHO intake, which would ensure an optimal consumption of CHO by offering variety in terms of food texture [3]. 

Overall, CHO intake by SOUT runners was below the updated recommendations regardless of whether a dietitian-nutritionist helped them during these events, although runners showed a preference for solid foods [5,29], also highlighting the consumption of gels and sports fluids [5,29,33]. For example, fructose consumption was represented by fresh or dehydrated fruits [5,29,33] without it being able to determine if the gels also contained fructose, since such information was not available in the relevant studies. Future studies may consider this type of content.

Linked to the concept of hydration, a suitable state during the race encourages the absorption of CHO [42] and prevents GI complications associated with the state of dehydration [26] caused by reduced blood flow to the digestive system. On the other hand, water is part of glycogen storage [58]. In four studies [3,5,8,33], faster runners reported higher fluid intake and recorded better performance, also matching an increase in CHO intake [5]. Hydration is an appropriate way to incorporate CHO [59], allowing an optimal state of hydration to be maintained [37] in order to avoid GI complications [16,20], which in turn entails disadvantages in the feeding plan and fluids designed by runners[6]. One strategy to ensure high CHO intake, avoiding GI complications associated with dehydration states, is through sports drinks or gels, ensuring suitable amounts of fluids in each of the latter [2,4].

On the other hand, one of the main drawbacks of ultra-endurance sports is the difficulty in reaching CHO intake recommendations [12]. Despite the benefits of high CHO/hour intake to maximize muscle glycogen reserves [12], delaying fatigue and improving post-exercise recovery [2,4], SOUT runners have shown an intake of 20–40 g CHO/h when such intake is “ad libitum” [37]. Only in two studies [2,4] did athletes consume the recommended quantity of (90 g/h) and the athletes achieved higher intake (120 g/h) due to the research protocol itself.

### 4.2. Fatigue Outcomes

The SOUT events involve metabolic [60] and physical [61] requirements. For this reason, a nutritional strategy must be ensured that allows recovery times to be shortened between sessions and competitions [2]. Moreover, these races have a prominent component of eccentric strength [45], whereby the runners are subjected to high levels of EIMD and fatigue [15].

#### 4.2.1. Internal Exercise Load

One of the review studies [2] focused the analysis on the effects of an intake of 120 g/h CHO on the internal load during a SOUT event. Different terrain in SOUT events involves eccentric and concentric muscle contractions [62,63], and these physical and metabolic requirements increase the internal exercise load [3,7], allowing the fatigue status of the athlete to be monitored [64]. The increase in internal load after completing training or competition [62] represents a reduction in performance. [2]. Urdampilleta et al. [2] explained how changes in muscle damage markers (CK, LDH, GOT) measured during a SOUT suggest that EIMD has an effect on internal exercise load and fatigue.

On the other hand, the storage and volume of muscle glycogen play a key role in regulating muscle function [65]. Excitation-contraction coupling constitutes the means of communication by which electrical events occur in the plasma membrane, including the release of Ca^2+^ from the sarcoplasmic reticle, resulting in muscle contraction [65]. Muscle glycogen levels affect this coupling, and so low levels of muscle glycogen cause a failure in coupling and consequently in the release of Ca^2+^ [65,66].

In summary, the increased internal load of exercise at SOUT events is associated with EIMD and muscle fatigue [2], which poses complications in muscle glycogen synthesis [67] and GLUT 4 translocation [68]. These complications arising from increased internal exercise load can be overcome via an intake of 120 g/h CHO during the event [2]. Such intake during the race allows reserves of muscle glycogen to be maintained and, alongside this, the metabolic and physiological balance [2,67].

This study quantified the physiological modifications that occur in SOUT, whereby the comparison between the various CHO intake groups made it possible to ascertain the benefits of an intake of 120 g/h CHO in the internal load. CHO intake is associated with central and peripheral fatigue [2,69], and so a suitable contribution improves the neuromuscular function [70]. In this sense, two studies [2,4] focused their analysis on the relevance of a suitable intake of CHO in order to reduce EIMD markers and internal exercise load.

#### 4.2.2. EIMD Markers

EIMD leads to multiple complications [71,72], namely muscle pain, loss of strength or impaired performance. Impaired muscle function compromises muscle glycogen resynthesis and GLUT4 translocation [68], resulting in a reduction in glucose transport to muscle cells. Alterations in the membrane determine insulin resistance after eccentric exercise is performed, as occurs during a SOUT, and lack of insulin leads to defective translocation of the GLUT4 protein in the membrane [73].

Scientific evidence has shown the relationship between CHO intake during SOUT and EIMD [2,4]. This relationship is based on the principle that damage to muscle cells involves a higher concentration of muscle proteins in circulation and thus leakage of extracellular [72] and intracellular [74] content as well as metabolic alterations (adenosine triphosphate (ADP), Pi, H+), in the release and absorption of Ca^2+^ from the sarcoplasmatic reticle [73]. This damage to muscle cells involves a difficulty in replenishing glycogen after prolonged physical exercise and/or high intensity [66]. Ørtenblad et al. [66] showed that intermyofibrillar glycogen content is related to the rate of release of Ca^2+^, and this deterioration implies less availability of glucose during recovery. Therefore, EIMD compromises muscle glycogen reserves [66] and GLUT4 translocation [68,73].

Likewise, EIMD supposes a deterioration of neuromuscular function and therefore an increase in muscle pain and proteins related to muscle metabolism [71,72], these conditions comprising muscle glycogen replacement. Thus, it has been demonstrated that it is possible to improve EIMD markers with an intake of 120 g CHO/h. The highest levels of these indicators (CK, LDH) usually appear after 24–72 h, requiring several days to return to reference values [24,25]. In the study by Viribay et al. [4] the blood markers of GOT, LDH, CK evidenced better values in the group that consumed 120 g CHO/h (HIGH) compared to groups with an intake of 90 or 60g CHO/h (CON and LOW respectively).

#### 4.2.3. Post-Exercise Recovery

Multiple factors should be taken into account at the time of post-exercise recovery, namely rehydration, repair of muscle damage, and removal of muscle glycogen [75,76,77]. In SOUT exercises, muscle glycogen depletion is one of the performance limiters [78], and glycogen replenishment plays a major part in recovering functional muscle capacity [75,79,80]. Having said this, athletes participating in SOUT can be considered for training with multiple sessions a day, at rates that hinder glycogen recovery times [75]. Therefore, when recovery is limited (<8 h), muscle glycogen cannot be fully restored [77], and for this reason, nutritional strategies should seek to accelerate glycogen resynthesis [76,77] and/or improve preconditions during the event in order to avoid ending up with low muscle glycogen reserves[2,4].

The study by Urdampilleta et al. [2] showed that 120 g/h intake of CHO ensures a permanent energy supply during the race so at the end, the energy deficit is lower. Athletes who compete or exercise in multiple sessions need suitable glycogen resynthesis capacity [2], which on the other hand is compromised by metabolic demand [29]. Thus, improving glycogen content during a SOUT allows for short and long-term recovery [2,4] without compromising the volume of sessions between it or the intensity of the mountain marathon.

Increasing the internal exercise load and EIMD compromises glycogen synthesis capability, which involves GLUT4 translocation [68] and excitation-contraction coupling failures [66].

In summary, two studies [2,4] included in this review demonstrated the benefits of ingesting an intake of 120 g/h CHO during a SOUT, because better levels of post-exercise glycogen slow down the onset of fatigue and prevent adverse effects of EIMD.

### 4.3. Effects of CHO Intake on GI Problems

Ultra-endurance physical exertion causes physical and psychological stress [11,81], which could lead to changes in the intestinal tract and consequently affect athletic performance [37]. GI problems are common, especially in endurance athletes, and often impair performance or subsequent recovery [82]. The main GI symptoms associated with running might also be explained by [34]: dietary factors, alterations in intestinal or colonic water, electrolyte fluxes, disturbances in GI motility or underlying irritable bowel syndrome [83]. Moreover, these alterations and those related to mechanism may be different in the case of the upper (stomach) and lower (intestine) regions [42], with major differences between them. One study into long-distance triathletes who competed in extreme conditions demonstrated a prevalence of up to 93% in any GI symptoms [84]. Similarly, a survey of 707 participants in the 13th Annual Trail’s End Marathon in Seaside, Oregon (USA), showed a high incidence of GI problems, predominantly of the lower tract [83]. Between 30% and 90% of participants experienced some type of these GI problems, making it one of the main causes of the deficit in the intake of CHO while competing or training [5,6,8,83]. In addition, GI symptoms are usually more frequent when they are related to permanent states of dehydration, associated pathologies or high levels of stress or anxiety [85] and are exacerbated by hot climate and elevated humidity conditions [27]. Moreover, beverages with high osmolality (>500 mOsm/L) seemed to be associated with an increased incidence of symptoms [34].

The main pathophysiological mechanism by which this GI occurs in athletes, especially in the gut, is the intestinal ischemia motivated by the release of norepinephrine that is stimulated by long-term exercise and the high intensity that induces a secondary vasoconstriction. This in turn produces an epithelial lesion, changes in GI permeability and disruption of the epithelial barrier function, increasing serum gastrin related to exercise and exacerbating low GI problems such as diarrhea—resulting in osmotic shifts [86]. Gastric emptying is regulated in the stomach by different mechanisms such as neural control, hormone control and the vagal motor circuit [87]. Exercise impact on all these mechanisms has not yet become well established to the author’s knowledge, although how acute exercise impacts some gastric “braking” hormones like glucagon-like peptide-1 (GLP), leptin and cholecystokinin (CCK) is documented in scientific literature, affecting gastric emptying when exercise intensity is increased [88,89].

If exercise continues or exercise intensity increases and dehydration worsens, the splanchnic blood flow decreases even more, and, together with continuous mechanical impact and increased sympathetic activation, GI symptoms are aggravated accordingly [8,90]. However, this process could be reduced through gut training, as argued by Costa et al. [8] and Jeukendrup, [42], since exposure to high CHO intake can increase gastric emptying and stomach comfort, thus reducing GI symptoms and optimizing the monosaccharide transport rate. This in turn is mediated by increased SGLT1 protein expression and, therefore, improves the oxidation rate. [91], being one of the most interesting types of gut training proposed by Jeukendrup [42].

Regarding the appearance of symptoms and running speed, the study by Costa et al. [38] observed that faster runners experienced 2.5 times more GI problems than slower ones during the race, which made it difficult for them to continue with the established nutritional intake plan [6]. Conversely, Costa et al. [8] concluded that faster runners were less affected by GI symptoms. This difference in profiles may be due to the fact that the faster ones had made a nutritional plan that consisted of a greater intake during high volumes of training prior to the competition—hence, the adaptation of the stomach and gut in order to facilitate the absorption of nutrients in stressful situations [8]. Specifically, the relative intensity of each athlete may be one of the factors that highlight the differences among studies in terms of GI symptoms. As such, despite the major prevalence of mild or severe symptoms, the etiology of these GI complaints in endurance athletes is still not completely understood [34].

On the other hand, Wardenaar et al. associated a greater total nutrient intake with the lower incidence of symptomatology (except for fiber), although they were unable to determine whether the lesser symptomatology was due to higher intake or whether a better choice in the type of food enabled a higher intake [6]. Possibly, this study associates high intake with lower GI symptoms, given that athletes who practice sport daily with a suitably usual intake assume that high CHO intake is part of such daily practice. Moreover, a greater intake of fluids and foods helps to maintain better hydration status and, therefore, limits GI problems [41]. Symptoms are considered more likely with high fiber intake, fats and highly concentrated CHO solutions (i.e., hypertonic drinks) [92], while hypertonic solutions encourage net water secretion into the intestinal lumen, resulting in a temporary net loss of water from the body [93]. Despite this, high intake may not be associated with greater symptoms as long as suitable intestinal training has been undertaken, which makes it possible to increase the ability to digest and absorb CHO via the optimum regulation of SGLT1, GLUT5 and GLUT2 transporters [42].

Only two studies [2,4] detail the type of CHO and ratio. In them, only three athletes manifested GI complications, and these were participants with prior intestinal training that allowed high intake of CHO (120 g/h) to be attained without GI complications. One study [8] found no link between CHO intake and intestinal symptomatology. On the other hand, in another study [5] there were few complications, or the high intake of CHO was associated with a low frequency of symptoms [6].

The information available in the studies does not allow whether or not the type of CHO intake had an influence on GI symptomatology to be determined. It should be considered that in some studies some participants deemed it optimal to consume CHO in foods such as sandwiches, while in others consumption focused on gels, those being the ones where there was also intestinal training. This intestinal strategy does not occur in other studies, and so it is not possible to determine whether CHO intake in liquid or solid form is an advantage in avoiding complications. Only two studies [3,5] allowed the source of CHO used by participants to be analyzed with detailed GI symptomatology. In one of them [3], the symptoms were not transcendent or impacted on the performance of the runners. Those who evidenced the most symptomatology were the ones who consumed solid foods—they do not specify which ones—followed by those who consumed sports drinks. It should be mentioned that there was a third category: soft foods (all foods that did not require chewing). Solid foods have greater involvement in the digestive system than liquids and consequently lead to a delay in gastric emptying [57], which will imply a reduction in the oxidation efficiency of CHO [57]. The other study [5] reported fewer symptoms during the race than afterward. The athlete who reported the most complications—urgency to defecate—coincided with an intake of chocolate milk and other solids and fluids other than gels, energy bars or sports drinks. It should be mentioned that another participant who experienced discomfort in the upper intestinal tract—flatulence—consumed cola. This case could be explained as referring to carbonated beverages. The lack of information does not allow a conclusion to be drawn as to in which case an intake of CHO from a solid, liquid or soft source corresponds to a major or minor GI complication, or which area of the intestinal tract would be most affected.

The appearance of GI symptoms is a limitation in SOUT events [5,6,8,30,32,38,94], being more frequent in runners with pre-existing symptoms, in slower athletes [34] and in situations where the state of dehydration state has intensified, such as in hot and humid environments [45]. There is a disparity among results related to speed and GI symptoms, possibly due to the different objectives and methodologies used in different studies. Overall, it seems that moderate exercise has little effect on GI tract motility, but when exercise is more severe, there may be some inhibiting effects, especially in terms of gastric emptying [34]. In some of them, the faster runners did not evidence any GI symptoms since they undertook intestinal training [2,4], while in other studies, the faster runners reported GI complications when trying to consume more CHO given that they failed to undertake bowel training [38]. Moreover, as Pfeiffer et al. [20] suggested, a correlation between individual predisposition and GI distress should be taken into account by understanding intra individual variability of these mechanisms, in order to prepare personalized and periodized nutrition for each athlete [17].

The amount of CHO is as important as the time when CHO is ingested, since a high amount of CHO, usually through a hyperosmolar solution in a short period of time, would involve GI complications [26]. Inadequate fluid intake makes it difficult to ensure gastric emptying [32], and in this case, it is advised that a 2:1 glucose–fructose ratio be maintained to attain high amounts of CHO ingestion [2].

## 5. Limitations and Strengths and Future Lines of Research

There is a small number of papers related to this topic that could constitute a limitation of this systematic review and that the results obtained from it should be viewed with caution. However, after reviewing some manuscripts that show the requirements for a high-quality systematic review, to the best of the authors’ knowledge, none of them refers to the minimum number of articles that should be included in it [93,94]. However, these authors speak of strict quality criteria when preparing a systematic review [93,94], and so it was carried out in accordance with PRISMA (Preferred Reporting Items for Systematic Reviews and Meta-analyses) statement guidelines and the PICOS model for the definition of inclusion criteria. Moreover, methodological quality and risk of bias were assessed by two authors independently, and disagreements were resolved by third party evaluation, in accordance with the Cochrane Collaboration Guidelines samples.

Additionally, there are some limitations related to the different protocols required for obtaining nutritional intake data, as well as performance measures and data associated with the GI problems of the studies included. One of the main topics that limits this systematic review is the comparison of the food eaten. In these studies, the disagreement with the methods used lay with collecting the intake during races. Another potential limitation is the fact that, in some studies, a team of researchers recorded the intake by athletes at checkpoints [6,8,9] or along the route [10]. However, in others, a self-reported questionnaire or reminder was used where athletes recorded everything consumed during the competition [6,29], and so, the intake described in these studies is not shown in g/h total grams [10]. Another major difference that can be seen as a potential limitation is the unit in which the information was reported. The recommendations are expressed in g of CHO/h [2,4,5,8,29], albeit in another three research studies in a g/kg/h ratio [6,29], which makes comparison between studies difficult.

In contrast, the main strength of this systematic review is the original idea of associating CHO intake with the three parameters (1: internal load 2: muscle damage and 3: GI symptoms), in an attempt to establish the main causes and consequences of low or high intake of CHO in these types of event.

Thus, future research should address nutritional strategies that allow athletes to consume amounts greater than 90 g of CHO/h in these races, particularly when environmental and/or geographical conditions (altitude, temperature, and humidity) are more difficult, given that these amounts evidence a positive effect on performance and a delay in the onset of fatigue and EIMD. Another line of research could be aimed at evaluating the relationship between GI symptoms and the source of carbohydrates, assuming that although sugars and starches are chemically the same, their physiological behavior is not the same if they are taken from the matrix of fruit or soft drink, or if they are naturally present in food or have been added to a recipe.

## 6. Practical Applications

Suitable periodization is relevant in order to understand the various challenges faced by SOUT better, including both physiological and metabolic demands and nutritional ones. A high intake of CHO during exercise would improve high intensity running capacity and short- and long-term recovery in SOUT competitions. As such, coaches and nutritionists should consider this nutritional strategy as a means to improve not only performance but also internal exercise load and delay in muscle fatigue.

Personalized nutrition should be based on athlete´s characteristics, needs and race-specific demands. For this reason, a previous nutritional examination is required to better understand the athlete´s preferences regarding foods and drinks used during practice sessions and competitions, as well as individual requirements in terms of nutrition and hydration. Previous logistic (feed zones, material and personal assistance) planning, as well as race profile and feed opportunity analysis, may be crucial in successfully completing the nutritional race strategy.

In order to guarantee high CHO intake during races, avoid GI problems and ensure stomach and intestinal comfort, previous intestinal training protocol implementation is required. Moreover, an individual maximum CHO intake threshold should be analyzed according to the athlete´s capacity to digest, absorb and oxidize it. Finally, it needs to be understood that eating over such a long time while running represents a huge challenge for athletes. Thus, both foods and beverage flavors and consistency should be previously tested and approved according to the athlete´s preferences, so as to try and vary them between sweet and salty tastes, as well as liquid, semi-solid and solid options.

## 7. Conclusions

Based on the results of this systematic review, most of the runners did not meet CHO intake standard recommendations for SOUT events (90 g/h). The main reasons why athletes referred to this complication regarding CHO intake were as follows: (1) the difficulty in eating all the required food or supplements and (2) the persistence of GI symptoms and/or appetite suppression.

Likewise, athletes who consumed more CHO evidenced a reduction in internal exercise load during SOUT events, even if such CHO intake did not adhere to recommendations. Similar benefits included intake of 120 g/CHO in muscle damage parameters observed in CK, LDH and GOT, with this intake evidencing benefits in short- and long-term recovery compared to an intake of 60 and 90 g/h CHO.

On the other hand, GI symptoms were recurrent in SOUT athletes depending on altitude, environmental conditions and running speed. Some overcame them via a nutritional strategy such as gut training, progressive consumption of CHO during practice or CHO intake, using different absorption transporters.

## Figures and Tables

**Figure 1 ijerph-18-05737-f001:**
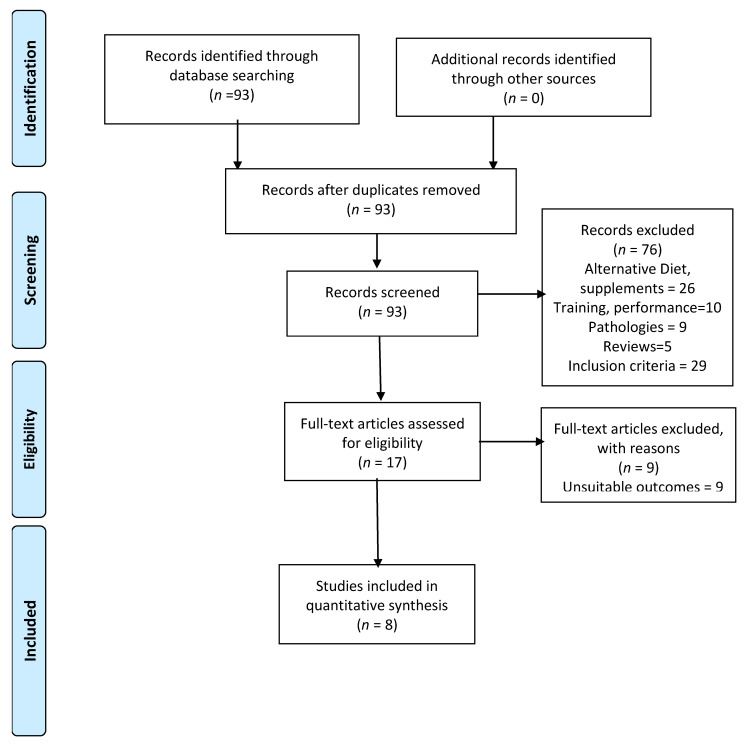
Study selection. Preferred Reporting Items for Systematic Reviews and Meta-Analyses (PRISMA) flow diagram.

**Table 1 ijerph-18-05737-t001:** Methodological quality and risk of bias of the studies included in the review.

Evaluated Items	Urdampilleta et al., 2020 [2]	Lavoué et al., 2020 [3]	Viribay et al., 2020 [4]	Wardenaar et al., 2018 [5]	Martínez et al., 2018 [27]	Wardenaar et al., 2015 [6]	Costa et al., 2014 [8]	Kruseman et al., 2005 [33]
Purpose	1	1	1	1	1	1	1	1
Literature Review	1	1	1	1	1	1	1	1
Study Design	1	0	1	1	1	1	1	1
Blinding	0	0	0	0	0	0	0	0
Sample Description	1	1	1	0	1	1	1	0
Sample Size	1	1	1	1	0	0	0	0
Ethics and Consent	1	1	1	1	1	1	1	1
Validity of Outcomes	1	1	1	1	0	1	1	1
Reliability of Outcomes	1	1	1	1	1	1	0	1
Intervention Description	1	1	1	1	1	1	1	1
Statistical Significance	1	1	1	0	1	1	0	0
Statistical Analysis	1	1	1	1	1	1	1	1
Clinical Importance	1	0	1	1	0	0	1	NR
Conclusions	1	1	1	1	1	1	1	1
Clinical Implications	1	0	1	1	0	0	1	0
Study Limitations	1	0	1	1	NR	NR	NR	1
TOTAL	12	8	12	13	10	12	11	11
%	75%	61.53%	75%	81.25	62.5	75.0	68.8	68.8
Methodological Quality	Good	Poor	Good	Very Good	Acceptable	Good	Good	Good

**Table 2 ijerph-18-05737-t002:** General characteristics of the studies included in the review.

Reference	Event	Distance/Elevation Gain Temperature	Participants	
			N (M + F) (Age)	Experience
Ultra-Trail				
Lavoué et al., 2020 [3]	Albi, France	24 h/ (distance range 133–272 km) 11.9–35.1 °C	11 (5 M + 6 F) (46 ± 7 years)	Elite
Wardenaar et al., 2018 [5]	Sixty of Texel, Netherlands	120 km/4.3–9.6 °C	5 M (47 ± 6 years)	Completed at least 10 (ultra) marathons
Martínez et al., 2018 [29]	Ultra Mallorca Serra de Tramuntana, Spain	112 km/ 4.448 m of ascent 11–21 °C	54 (53 M + 1 F) (35.7 ± 7.9 years)	Data not shown
		67 km/2521 m of ascent 11–21 °C	109 (98 M + 11 F) (35.2 ± 8.4 years)	
Wardenaar et al., 2015 [6]	Sixty of Texel, Netherlands	120 km/2.2 °C	8 (7 M + 1 F) (46.6 ± 6.3 years)	Data not shown
		60 km/2.2 °C	60 (48 M + 12 F) (46.5 ± 7.2 years)	
Costa et al., 2014 [8]	Glenmore 24 Trail Race, Scottish Highlands, Scotland, UK	24 h/(distance range: 122–208 km) 0–20 °C	25 (19 M + 6 F) (39 ± 7 years)	Data not shown
Kruseman et al., 2005 [33]	Tour des Dents du Midi race, Switzerland	44 km/2890 m of ascent 18–30 °C	42 (39 M + 3 F) (42 ± 9.7 years)	Completed at least 10 trail marathons
Trail Marathon				
Urdampilleta et al., 2020 [2]	Oiartzun, Spain	10 °C/Cumulative elevation gain: 3980.80 m	20 M LOW: 37.8 ± 9.4 years CON: 37.2 ± 5.4 years EXP: 38 ± 6.8 years	5 years in SOUT (2 World champions)
Viribay et al., 2020 [4]	Oiartzun, Spain	10 °C/Cumulative elevation gain: 3980.80 m	20 M LOW: 37.8 ± 9.4 years CON: 37.2 ± 5.4 years EXP: 38 ± 6.8 years	5 years in SOUT (2 World champions)
Martínez et al., 2018 [29]	Ultra Mallorca Serra de Tramuntana, Spain	11.0–21.6 °C/1424 m of ascent	63 (51 M + 12 F) (36.6 ± 8 years)	Unshown data

M: males; F: females; LOW: 60 g/h of carbohydrate intake during the marathon; CON: 90 g/h of carbohydrate intake during the marathon; EXP: 120 g/h of carbohydrate intake during the marathon; #: average.

**Table 3 ijerph-18-05737-t003:** Carbohydrate (CHO) amount ingested during SOUT events.

Reference	Fluid Intake	CHO Intake	Type of CHO	Vs. Recommendations (90 g/h)	Nutritional Strategies	Observations
Ultra-Trail
Lavoué et al., 2020 [3]	274 ± 115 mL/h	13.9–105.4 g/h (62.2 ± 29.6 g/h)	Sports drink, cake, fruit, and mashed potatoes.	↓	YES	Higher rates of energy intake for finishers relative to those of non-finishers and for fast runners compared to slow runners
685 ± 290 mL/kg BM
Wardenaar et al., 2018 [5]	354–765 mL/h	46.5 ± 14.1 g/h (range: 22.1–62.6 g/h)	Fruit, gels, sports drink, chocolate milk	↓	NO	↑ CHO intake in the section 75–90 km because ↓lower running speed
Martínez et al. 2018 [29]	6.319 ± 4214 L	(a) 112 km: 534.9 ± 279.3 g/total (31.2 ± 17.8 g/h)	(a) & (b) Sandwiches, fruit (mainly bananas), gels, pasta, energy bars CHO-electrolyte drinks	(a) ↓	NO	No difference among distances. The slow paces during race could mean that participants did not require as much CHO.
351 ± 239 mL/h	(b) 67 km: 326.7 ± 157.2 g/total (32.1 ± 14.8 g/h)	(b) ↓
Wardenaar et al., 2015 [6]	2.9 ± 0.9 L	(a) 60 km: 274 ± 133 g/total	No data shown	(a) ↓	NO	(a) 22% of runners kept to CHO recommendations (21.2% males and 12.5% females)
(b) 120 km: 67.3 ± 31.7 g/h	(b) ↓	(b) Only 1 runner kept to CHO recommendations
Costa et al., 2014 [8]	9.1 ± 4.0 L	(a) <160 km: 31 ± 9 g/h	Mono/di/oligosaccharide, polysaccharide sources	(a) ↓	NO	CHO rates ranged from 16 to 53 g/h (only 1 runner of (≥160 km) consumed 126 g/h).
118 ± 51 mL/kg BM	(b) ≥160 km: 44 ± 33 g/h	(b) ↓
Kruseman, et al., 2005 [33]	3.777 ± 1.146 L	31 ± 14 g/h	Sweet drinks and glucose. Slowest: soup, fruits and cereal bars. Fastest: Gels	↓	NO	>50% runners: <30 g/h 3 runners = 60 g/h insufficient palatability of fluid and food, the practical difficulty of drinking or eating while running/walking remains possible
545 ± 158 mL/h
Trail Marathon
Urdampilleta et al., 2020 [2]	No data shown	(a) LOW: 60 g/h	30 g/h maltodextrin (glucose) and fructose gels (ratio 2:1)	(a) ↓	YES	↑ CHO intake decreasing internal exercise load and neuromuscular fatigue
(b) CON: 90 g/h	(b) 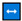
(c) EXP: 120g/h	(c) ↑
Viribay et al., 2020 [4]	No data shown	(a) LOW: 60 g/h	30 g/h maltodextrin (glucose) and fructose gels (ratio 2:1)	(a) ↓	YES	↑ CHO intake decreasing internal exercise load and EIMD.
(b) CON: 90 g/h	(b) 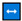
(c) EXP: 120g/h	(c) ↑
Martínez et al. 2018 [29]	4.727 ± 2694 L	205.2 ± 81.2 g (33.4 ± 13.5 g/h)	Sandwiches, fruit (mainly bananas), gels, pasta, energy bar, CHO-electrolyte drinks	↓	NO	No difference between distances. The slow paces during the race could mean that participants did not require as much CHO
459 ± 256 mL/h

CHO: carbohydrates; EIMD: exercise induced muscle damage LOW: 60 g/h of carbohydrate intake during marathon; CON: 90 g/h of carbohydrate intake during marathon; EXP: 120 g/h of carbohydrate intake during marathon; ↓: lower; 
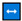
: equal; ↑: Higher.

**Table 4 ijerph-18-05737-t004:** Summary of studies included in the systematic review that researched into the effect of carbohydrates on fatigue (internal exercise load, EIMD markers and post-exercise recovery).

**Author**	**Fatigue Related Outcomes Studied**	**Conclusions**
Internal exercise load		
Urdampilleta et al., 2020 [2]	TRIMP	↓ HIGH vs. LOW/MED
Viribay et al., 2020 [4]	Session-RPE method	↓ EXP vs. LOW/CON
EIMD markers		
Viribay et al., 2020 [4]	GOT LDH CK	↓ EXP vs. LOW/MED ↓ EXP vs. LOW/MED ↓ EXP vs. LOW/MED
Post-exercise recovery		
Urdampilleta et al., 2020 [2]	ABK_JT_/ABK_H_: HST_1-RM_/ HST_Speed_: Borg: ABK_H_: HST_1-RM_: HST_Seepd_: High Intensity Run Capacity (better time)	↓ HIGH vs. LOW/MED ↓* HIGH vs. LOW/MED ↓ HIGH vs. LOW/MED ↑ HIGH vs. LOW/MED ↓* HIGH vs. LOW/MED ↓* HIGH vs. LOW/MED ↑ HIGH vs. LOW/MED

↓: decrease; ↓*: smaller loss; ↑: improvement; HIGH or EXP: 120 g/h; MED or CON: 90 g/h; LOW: 60 g/h; GOT: glutamic oxaloacetic transaminase; LDH: lactate dehydrogenase; CK: creatine kinase; ABK_JT_: Abalakov jump time; ABK_H_: Abalakov jump height; HST_Speed_: speed of half-squad test; HST_1-RM_: 1-repetition maximum test of half squat test.

**Table 5 ijerph-18-05737-t005:** Summary of studies included in the systematic review that researched into the effect of carbohydrates (CHO) on gastrointestinal (GI) symptoms during exercise.

Autor	Upper GI Symptoms	Lower GI Symptoms	Conclusions
Ultra-Trail			
Lavoué et al., 2020 [3]	8 participants experienced at least one GI (nausea = 4, difficulty swallowing = 3 vomiting = 1)	Diarrhea (*N* = 2)	The episodes of GI were transient and did not cause any major decreases in performance or dropping out
Wardenaar et al., 2018 [5]	Nausea (*N* = 1)	Urge to defecate (*N* = 3) Flatulence (*N* = 3) Side (*N* = 1)	Lower amount of GI complaints during the race than the post-race
Wardenaar et al., 2015 [6]	Reflux (16.3%) Heartburn (9.3%) Belching (41.9%) Bloating (16.3%) Stomach cramps (14.0%) Nausea (20.9%)	Intestinal cramp (9.3%) Flatulence (34.9%) Urge to defecate (16.3%) Side ache (11.6%) Abdominal pain (9.3%) Loose stool (4.6%) Diarrhea (2.3%)	Higher nutrient intake, except fiber intake, was in general associated with lower frequency of GI distress
Costa et al., 2014 [8]	65% reporting at least one severe symptom (nausea; GI pain; vomiting; indigestion; bloating; abnormal bowel movements (e.g., urgency to defecate)		No association between GI and energy and CHO intake was evident. A 2·5-fold greater occurrence of Gl symptoms was observed in the fast group vs. slow group
Trail Marathon			
Urdampilleta et al., 2020 [2]	3 with flatulence and/ or reflux		Athletes with gut training did not evidence any GI problem
Viribay et al., 2020 [4]	3 with flatulence and/ or reflux		Athletes with gut training did not evidence any GI problem

Upper GI complaints such as: reflux, heartburn, belching, bloating, stomach cramps, vomiting and nausea; lower GI, such as: intestinal cramp, flatulence, urge to defecate, side ache, abdominal pain, loose stool, diarrhea, and intestinal bleeding. Other, dizziness, headache, muscle cramps, urge to urinate.

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
