# Peer review of "Relationship of Carbohydrate Intake during a Single-Stage One-Day Ultra-Trail Race with Fatigue Outcomes and Gastrointestinal Problems: A Systematic Review"

_ijerph, 2021, doi:10.3390/ijerph18115737_

Round 1

Reviewer 1 Report

Dear authors,

Authors based on the increased number of participants on one-day ultra-trail (SOUT) race have performed a systematic review with a practical applications aims. It´s evident that the ultra-trail runners must optimize the nutrition during the competition. In this sense, it´s very useful to optimize the dose of carbohydrate (CHO) ingestion during competition. Nevertheless, the systematic review presents some difficulties and numerous deficiencies that make impossible to publish it.

The first difficult is to consider compare the CHO ingestion isolated and don´t consider the hydric ingestion and the dehydration rate of the athletes. The hydric status and the dehydration rate affect to the absorption rate of CHO. During the systematic review, it cannot be differentiated the effect of CHO ingestion in conditions with a similar hydration status. This has been a very important limitation of the systematic review that affect to the reliability of it.

In addition, authors during the systematic review didn´t compare the effect of different type of CHO in composition and format. What relation exist between the gastrointestinal problems (GI) and different dietary sources (sport bar or sport drinks or sandwiches)? It´s impossible to finish a systematic review about this topic and cannot be differentiate the specific effect of different dietary source enriched on CHO.

Other important limitation it´s the definition of GI. What´s a GI on the review? What instruments exist and what it´s the reliability of the used on the studies of this systematic review? Is it possible to study all the possible GI as one?

These important methodological limitations suggest to reject this systematic review.  

Author Response

Point-by-Point Response to Reviewer’s Comments

We would like to sincerely thank the reviewer for his/her helpful recommendations. We have seriously considered all your comments and carefully revised the manuscript accordingly. Revisions are highlighted in yellow through the manuscript to indicate where changes have taken place. We feel that the quality of the manuscript has been significantly improved with these modifications and improvements based on the reviewers’ suggestions and comments. We hope our revision will lead to an acceptance of our manuscript for publication in Nutrients.

In advanced

Kind regards

Reviewer 1

Dear authors,

Authors based on the increased number of participants on one-day ultra-trail (SOUT) race have performed a systematic review with a practical applications aims. It´s evident that the ultra-trail runners must optimize the nutrition during the competition. In this sense, it´s very useful to optimize the dose of carbohydrate ingestion during competition. Nevertheless, the systematic review presents some difficulties and numerous deficiencies that make impossible to publish it.

Review Report 

REVIEWER: The first difficult is to consider compare the CHO ingestion isolated and don´t consider the hydric ingestion and the dehydration rate of the athletes. The hydric status and the dehydration rate affect to the absorption rate of CHO. During the systematic review, it cannot be differentiated the effect of CHO ingestion in conditions with a similar hydration status. This has been a very important limitation of the systematic review that affect to the reliability of it.

AUTHORS: We really appreciate your contribution to our scientific article. This is the main reason why we have included the hydration data in Table 3 and the corresponding development in the results section. Similarly, we have included the relevance of hydration status in the storage of CHOs and its influence on GI complications in cases of deficient hydration. We hope that with both changes we will respond to the comment, as a limitation of the study.

REVIEWER: In addition, authors during the systematic review didn´t compare the effect of different type of CHO in composition and format. What relation exist between the gastrointestinal problems (GI) and different dietary sources (sport bar or sport drinks or sandwiches)? It´s impossible to finish a systematic review about this topic and cannot be differentiate the specific effect of different dietary source enriched on CHO.

AUTHORS: We appreciate your feedback. Regarding to this point, it is complex to establish conclusively what is the most optimal type and form of CHO for consumption, given that each study presents differences. In its regard, all of them are very heterogeneous. This means that in some studies some athletes opted for solid or liquid foods (the studies describe ad libitum), while two of them, being experimental, restricted the options in relation to the form of CHO. In this sense, there is no more scientific information published about it for our knowledge. Despite, we have included how science has established, the best option for a high intake of CHO without causing an aggravation of GI symptoms, assuming the limitations of this type of studies with little information described in this regard, but that the strength relates the final state of current knowledge.

REVIEWER: Other important limitation it´s the definition of GI. What´s a GI on the review? What instruments exist and what it´s the reliability of the used on the studies of this systematic review? Is it possible to study all the possible GI as one?

AUTHORS: Thanks for the detail, you are right. We have incorporated what is considered as GI symptoms and how the studies classified the different symptoms, one of them based on ultra-endurance races of Nikolaidis et al., (2018) and regarding the GI, a reference study by Jeukendrup (2017). Among the studies that incorporated GI symptoms, two of them established a questionnaire and a scale with symptoms to assess symptoms. In this context, we have included next paragraph to explain these issues: “Besides, in this context, medical complications related to dietary-nutritional intake such as gastrointestinal (GI) problems, hyponatremia, dehydration, appetite suppression, as well as the difficulty of carrying the food and/or supplements are evident. These are the main reasons that adequate nutrition [17] is a key factor for optimal performance [11].”

Nikolaidis, P.T.; Veniamakis, E.; Rosemann, T. Nutrition in Ultra-Endurance : State of the Art. Nutrients 2018, 10, 1995.

Jeukendrup, A.E. Periodized Nutrition for Athletes. Sport. Med. 2017, 47, 51–63.

Reviewer 2 Report

This manuscript provides a systematic review of the literature on the relation of carbohydrate intake during a single-stage one-day ultra-trail race on fatigue and gastrointestinal problems.

Major comments

Upon reading this manuscript, I see that the introduction is poorly written and needs to be rewritten by someone who has better English writing skills. Please consider employing a professional writer to edit and rewrite this section of the manuscript.

What do you mean by "different distances can be finished"? Does this mean that competitors run for a predetermined time and attempt to run as far as possible within the given time period? Pleas add clarity to this sentence.

Minor comments

Line numbers are missing from the early pages.

Page numbers go back to page 2 after page 9.

Abstract line 3: Change "highly" to "high"

Abstract line 5: delete "the" before "fatigue"

Abstract line 8: Change "until" to "up to"

Abstract line 12: Change "runners who intake 120 g/h could..." to "runners, whose intake was 120 g/h, could..."

Abstract line 12: Change "limit the EIMD" to "limiting EIMD"

Abstract line 14: Change "run" to running"

Abstract line 15: Change "range GI symptoms" to "range of GI symptoms"

Abstract line 15: Do you mean the number of athletes with symptoms was 65-82%? Please rewrite this sentence to add clarity.

Abstract line 17: Change "showed decreasing in internal exercise load" to "a reduced internal exercise load"

Abstract line 17: Change "limit" to "limited"

Abstract line 18: Change "On the other hand" to "Conversely"

Abstract line 20: This final sentence is incomplete and has grammatical errors. Please rewrite the sentence so that the reader can understand the meaning.

Introduction

Line 1: Change to "Participation in single-stage one-day ultra-trail (SOUT) races has increased in recent years.

Line 2: Change to These events are defined as competitions lasting..."

Page 2, line 3: Change "events" to "event" and define "internal load."

Explain how EIMD affects performance. You need to make the distinction between EIMD and fatigue.

Page 2, line 5: Change "abandonment participation" to "failure to complete the event."

Page 2, line 8-9: Change to "These are the main reasons that adequate nutrition is a key factor for optimal performance."

Page 2, line 10: Delete "Concretely," Start the sentence with "In prolonged strenuous exercise..."

Page 2, line 11: Change "during exercise..." to "during the event..." Change "In this line," to "Consequently,"

Page 2, paragraph 3, line 2: Change consume smaller" to "consume s smaller" Also, smaller than what?

 Page 2, paragraph 3, line 8: Delete "In this sense."

Page 2, paragraph 3, line 11: Change "contain" to "contained"

Page 2, paragraph 4, line 2: Change "for the best..." to "to the best..."

Page 2, paragraph 4, line 3-4: You said "...relationship of CHO intake with...characteristics of CHO intake" The meaning here is unclear.

Methods

This section is generally clearly written with few grammatical errors. 

The systematic review protocol used is appropriate and thorough.

Inclusion and exclusion criteria are appropriate for this systematic review.

Outcome measures and quality assessment of the experiments sections are clear.

Tables in the Methods and Results sections are all clear and helpful for the reader.

Results

Figure 1 is clear and helpful for the reader.

Discussion

Line 32: Change "synthetize" to "synthesize" 

Line 36: Change "update" to "updated" or "latest" or "up-to-date"

Line 39: Delete "the"

Line 46-47: Change "stores places" to " "storage areas"

Line 53: Change "Besides" to "Importantly"

Line 55: Change "On the other hand" to " Conversely"

Line 60: Change "comparing with" to "compared with"

Line 73: Change "In this line" to "Accordingly"

Line 79: Change "the usual intake of training" to "the same as in training" 

Line 147: Delete "In this line."

Line 190: Change "of the review" to "included in this review

The writing is much better in the Discussion section than the Introduction. However, there are still basic errors that must be addressed.

The conclusions are valid based on the findings of the studies included in this systematic review.

Author Response

Point-by-Point Response to Reviewer’s Comments

We would like to sincerely thank the reviewer for his/her helpful recommendations. We have seriously considered all the comments and carefully revised the manuscript accordingly. Revisions are highlighted in yellow through the manuscript to indicate where changes have taken place. We feel that the quality of the manuscript has been significantly improved with these modifications and improvements based on the reviewers’ suggestions and comments. We hope our revision will lead to an acceptance of our manuscript for publication in Nutrients.

In advanced

Kind regards

Reviewer 2

This manuscript provides a systematic review of the literature on the relation of carbohydrate intake during a single-stage one-day ultra-trail race on fatigue and gastrointestinal problems.

Major comments

REVIEWER: Upon reading this manuscript, I see that the introduction is poorly written and needs to be rewritten by someone who has better English writing skills. Please consider employing a professional writer to edit and rewrite this section of the manuscript.

AUTHORS: Thank you for your recommendation. The authors have sent the manuscript to review better the English.

REVIEWER: ¿What do you mean by “different distances can be finished”? Does this mean that competitors run for a predetermined time and attempt to run as far as possible within the given time period? Pleas add clarity to this sentence.

AUTHORS: Thank you for your interest. That sentence describes that in one study it ran for time and not for distance, i.e. how much they could run in 24 hours? The corresponding clarification was incorporated in the test: “These events (ultra-endurance races) are defined as competitions lasting more than 4 hours and less than 24 h, where each participant must run for 24 hours as many kilometers as possible.”

REVIEWER: Abstract line 3: Change "highly" to "high"

AUTHORS: Revised as requested.

REVIEWER: Abstract line 5: delete "the" before "fatigue"

AUTHORS: Revised as requested.

REVIEWER: Abstract line 8: Change "until" to "up to"

AUTHORS: Revised as requested.

REVIEWER: Abstract line 12: Change "runners who intake 120 g/h could..." to "runners, whose intake was 120 g/h, could..."

AUTHORS: Revised as requested.

REVIEWER: Abstract line 12: Change "limit the EIMD" to "limiting EIMD"

AUTHORS: Revised as requested.

REVIEWER: Abstract line 14: Change "run" to running"

AUTHORS: Revised as requested.

REVIEWER: Abstract line 15: Change "range GI symptoms" to "range of GI symptoms"

AUTHORS: Revised as requested.

REVIEWER: Abstract line 15: Do you mean the number of athletes with symptoms was 65-82%? Please rewrite this sentence to add clarity.

AUTHORS: Thank you for your comment. The authors have rewritten the sentence: In 6 studies, athletes had GI symptoms between 65-82%.

REVIEWER: Abstract line 17: Change "showed decreasing in internal exercise load" to "a reduced internal exercise load"

AUTHORS: Revised as requested.

REVIEWER: Abstract line 17: Change "limit" to "limited"

AUTHORS: Revised as requested.

REVIEWER: Abstract line 18: Change "On the other hand" to "Conversely"

AUTHORS: Revised as requested.

REVIEWER: Abstract line 20: This final sentence is incomplete and has grammatical errors. Please rewrite the sentence so that the reader can understand the meaning.

AUTHORS: Thank you for your comment. The authors have rewritten the sentence: Therefore, a high CHO intake during SOUT events is important to delay fatigue and avoid GI complications. To ensure high intakes, it is necessary to implement intestinal training protocols.

Introduction

REVIEWER: Line 1: Change to "Participation in single-stage one-day ultra-trail (SOUT) races has increased in recent years.

AUTHORS: Revised as requested.

REVIEWER: Line 2: Change to These events are defined as competitions lasting..."

AUTHORS: Revised as requested.

REVIEWER: Page 2, line 3: Change "events" to "event" and define "internal load." Explain how EIMD affects performance. You need to make the distinction between EIMD and fatigue.

AUTHORS: Thank you for your recommendation. The authors have written the sentence:

Fatigue is classified as proximal (central fatigue) or distal (peripheral fatigue) to the neuromuscular junction. The first implies a reduction in activation, it reflects the inability to recruit all the motor units needed to maximize strength. On the other hand, peripheral fatigue could affect three main components: (1) action potential transmission along sarcolemma, (2) excitation – contraction coupling, and (3) actin-myosin interaction. The fatigue generated in this type of events and its impact in the internal load- psychophysiological response that occurs during exercise- during the race, is associated with decrease in glycogen stores, therefore exercise induce muscle damage (EIMD). The EIMD could affect performance by temporary decrease in muscle strength, increased muscle pain and swelling, and an increase in intramuscular proteins in the blood and even lead to failure to complete the event [8,14]. Besides, in this context, medical complications related to dietary-nutritional intake such as gastrointestinal (GI) problems, hyponatremia, dehydration, appetite suppression, as well as the difficulty of carrying the food and/or supplements are evident. These are the main reasons that adequate nutrition [15] is a key factor for optimal performance [11].

REVIEWER: Page 2, line 5: Change "abandonment participation" to "failure to complete the event."

AUTHORS: Revised as requested.

REVIEWER: Page 2, line 8-9: Change to "These are the main reasons that adequate nutrition is a key factor for optimal performance."

AUTHORS: Revised as requested.

REVIEWER: Page 2, line 10: Delete "Concretely," Start the sentence with "In prolonged strenuous exercise..."

AUTHORS: Revised as requested.

REVIEWER: Page 2, line 11: Change "during exercise..." to "during the event..." Change "In this line," to "Consequently,"

AUTHORS: Revised as requested.

REVIEWER: Page 2, paragraph 3, line 2: Change consume smaller" to "consumes smaller" Also, smaller than what?

AUTHORS: It refers to the lower consumption of CHO compared to the recommendations. Final writing: Although the CHO intake recommendations, most ultra-endurance athletes consume smaller amounts of CHO, compared to the macronutrient recommendations during SOUT competitions.

REVIEWER: Page 2, paragraph 3, line 8: Delete "In this sense."

AUTHORS: Revised as requested.

REVIEWER: Page 2, paragraph 3, line 11: Change "contain" to "contained"

AUTHORS: Revised as requested.

REVIEWER: Page 2, paragraph 4, line 2: Change "for the best..." to "to the best..."

AUTHORS: Revised as requested.

REVIEWER: Page 2, paragraph 4, line 3-4: You said "...relationship of CHO intake with...characteristics of CHO intake" The meaning here is unclear.

AUTHORS: Final writing: Although some extensive reviews provide general information about CHO intake and its relationship with GI symptoms [35,47], to the best of the authors' knowledge, none of them explored in detail the relationship of CHO intake with fatigue parameters and GI problems.

Discussion

REVIEWER: Line 32: Change "synthetize" to "synthesize" 

AUTHORS: Revised as requested.

REVIEWER: Line 36: Change "update" to "updated" or "latest" or "up-to-date"

AUTHORS: Revised as requested.

REVIEWER: Line 39: Delete "the"

AUTHORS: Revised as requested.

REVIEWER: Line 46-47: Change "stores places" to " "storage areas"

AUTHORS: Revised as requested.

REVIEWER: Line 53: Change "Besides" to "Importantly"

AUTHORS: Revised as requested.

REVIEWER: Line 55: Change "On the other hand" to " Conversely"

AUTHORS: Revised as requested.

REVIEWER: Line 60: Change "comparing with" to "compared with"

AUTHORS: Revised as requested.

REVIEWER: Line 73: Change "In this line" to "Accordingly"

AUTHORS: Revised as requested.

REVIEWER: Line 79: Change "the usual intake of training" to "the same as in training" 

AUTHORS: Revised as requested.

REVIEWER: Line 147: Delete "In this line."

AUTHORS: Revised as requested.

REVIEWER: Line 190: Change "of the review" to "included in this review

AUTHORS: Revised as requested.

REVIEWER: The writing is much better in the Discussion section than the Introduction. However, there are still basic errors that must be addressed.

AUTHORS: Thanks so much for this detail. We have rewritten the introduction part to delete these mistakes.

REVIEWER: The conclusions are valid based on the findings of the studies included in this systematic review.

AUTHORS: Thanks so much for your last comment and positive feedback. On the other hand, thanks one more time for your effort and comments in order to improve the final version of our document. We have considered all your suggestions.

Reviewer 3 Report

Congratulations to the authors for their work.
As a suggestion, it would have been interesting to study the relationship between GI symptoms and the source of carbohydrates, assuming that although sugars and starches are chemically the same, their physiological behavior is not the same if they are taken from the matrix of a fruit or soft drink or if they are naturally present in food or have been added to a recipe.

Author Response

Point-by-Point Response to Reviewer’s Comments

We would like to sincerely thank the reviewer for his/her helpful recommendations. We have seriously considered all the comments and carefully revised the manuscript accordingly. Revisions are highlighted in yellow through the manuscript to indicate where changes have taken place. We feel that the quality of the manuscript has been significantly improved with these modifications and improvements based on the reviewers’ suggestions and comments. We hope our revision will lead to an acceptance of our manuscript for publication in Nutrients.

In advanced

Kind regards

Reviewer 3

Congratulations to the authors for their work.

REVIEWER: As a suggestion, it would have been interesting to study the relationship between GI symptoms and the source of carbohydrates, assuming that although sugars and starches are chemically the same, their physiological behavior is not the same if they are taken from the matrix of a fruit or soft drink or if they are naturally present in food or have been added to a recipe.

AUTHORS: We thank you for your congratulations and feedback to improve the article. We have incorporated the analysis by crossing the information of the type of food consumption and symptomatology, whenever the article provided the necessary information to cross both data. For example, one of the items (Martínez et al., 2018) that provides more detail about the type of food consumed- fruit-sandwich-gel, gummies, energy bars- does not provide information on GI symptoms. On the other hand, another study (costa et al., 2014) did not specify the food source but gave data on the type of CHO: /mono/di/oligosaccharide or polysaccharide.

In this sense, given that we find this reflection made by the reviewer interesting, and considering that we cannot answer it because the studies do not relate these two variables, we have included this issue in future lines of research section, as a potential research work with this population: “Other research line could be aimed at evaluating the relationship between GI symptoms and the source of carbohydrates, assuming that although sugars and starches are chemically the same, their physiological behavior is not the same if they are taken from the matrix of a fruit or soft drink or if they are naturally present in food or have been added to a recipe.”

Round 2

Reviewer 2 Report

This manuscript provides a review of the literature that investigates carbohydrate intake during single-stage one-day ultra-trail racing and its effect on fatigue and gastrointestinal problems.

While the authors have made several changes to improve the manuscript, it appears they have changed only the examples provided by the reviewers. There are still several sections throughout the manuscript that are unclear due to poor sentence structure, It would be very helpful to have a native speaker of English help with revising the manuscript as the content appears to be sound.

For example the first paragraph in section 3.2 that describes the studies selected for inclusion in the review is very confusing.

Also, in the next paragraph, "of" is often used when "for" should be used. While, individually, these are very minor points, when these occur throughout much of the manuscript, it does become laborious to read.

I suggest one more edit by a native speaker before it can be accepted for publication.

Author Response

Response to Reviewer 2 suggestion

We would like to sincerely thank the reviewer for his/her helpful suggestion again. The manuscript has been linguistically edited by Philip Cooper, a British native professional translator and regular collaborator for the translation and revision of scientific articles at the University of Deusto (please see the attached English editing certificate). Revisions are highlighted in track changes through the manuscript in order to indicate where changes have taken place. We feel that the quality of the manuscript has been significantly improved with these modifications and improvements based on your recommendations. We hope our revision will lead to an acceptance of our manuscript for publication in International Journal of Environmental Research and Public Health.

In advanced

Kind regards
